# A physiological and histological atlas of reproduction in the North American deer mouse (*Peromyscus maniculatus*)

Kathryn Wilsterman[1,2]*, Megan J. Hemmerlein[1,2], Anna Isabel Bautista[1,3], Natalie M. Báez-Torres[4], Kaylinn M. Gosney[1], Kylie E. Jewett[1], Ashley M. Larson[1], Ellery L. Myers[1]

1 Department of Biology, Colorado State University, Fort Collins, Colorado, United States of America, 2 Cellular and Molecular Biology Graduate Program, Colorado State University, Fort Collins, Colorado, United States of America, 3 Department of Psychology, Emory University, Atlanta, Georgia, United States of America, 4 Division of Sciences and Technology, Universidad Ana G. Méndez-Cupey Campus, San Juan, PR, United States of America

* k.wilsterman@colostate.edu (KW)

## Abstract

The North American deer mouse (*Peromyscus maniculatus*) exhibits extensive diversity in morphology, physiology, and life history across its broad range. These traits have propelled the deer mouse to model system status across several fields within the biological sciences. Nonetheless, we still lack basic knowledge about some important aspects of this species' biology. For example, limited information about the deer mouse's reproductive physiology remains a significant barrier to developing genetic tools for the species and for advancing our current understanding of how this species has been so evolutionarily successful. Here, we aim to fill this knowledge gap by (1) characterizing body temperature profiles across reproductive stages and (2) generating a detailed histological atlas of placental development. We show that body temperature can be used to diagnose copulation and pregnancy in deer mice, however body temperature cannot be used to predict fertility (likelihood to breed) prior to pairing individuals. Our histological atlas of placental development represents the first day-by-day developmental timeline of the placenta in a *Peromyscus* species; using this atlas, we describe unique organization and behaviors of trophoblast cells in the deer mouse. Together, these descriptive datasets provide substantial new comparative data on reproductive physiology in Cricetids, and they provide a foundation for further functional work in this important model species.

## Introduction

The North American Deer Mouse (*Peromyscus maniculatus*; hereafter, 'deer mouse') is the most wide-spread small mammal in North America, occurring from northern Canada down through southern Mexico and from elevations as low as the basin of

**Data availability statement:** The data and scripts underlying the results presented in the study are available at https://github.com/mjhemmerlein/Peromyscus-Gestational-Atlas. Digital copies of images used to generate data can be found in Dryad using DOI: 10.5061/dryad.z34tmpgr7.

**Funding:** This research was supported by fellowship support for MJH from T32GM132057, an award from the Boettcher Foundation's Webb-Waring Biomedical Research Awards program to KW, and NIH R01 HD114615 to KW.

**Competing interests:** The authors have declared that no competing interests exist.

Death Valley (86 m below sea level) to the peaks of the Rockies (>4000 m) [1,2]. Across this broad distribution, deer mice display a remarkable capacity to morphologically and physiologically adapt to their local environments [1]; the abundance of natural diversity across deer mouse populations and their amenability to lab-based experimental research has even earned them the title of "The Drosophila of North American Mammalogy" [2]. In line with this namesake, deer mice have emerged as a prominent mammalian model system, particularly for evolutionary biology, over the past 30 years [3–10].

Deer mice are also an increasingly important model system for fields beyond evolutionary biology, including community ecology and disease ecology. Most notably, deer mice serve as reservoirs or host species for several important diseases, including tick-borne diseases and Hantaviruses [11,12], making them a species of interest for disease modeling and ecology. More recently, deer mice have also been identified as a useful model for studying the biology and virulence of emerging diseases like Coronaviruses, including Covid-19 [11]. The relevance and tractability of deer mice for understanding ecological and immunological processes further underscore the importance of this model mammal across biology.

Despite the deer mouse's broad utility as a model, there remain some major gaps in our knowledge of their basic biology. One of the most prominent obstacles is our general lack of knowledge about their reproductive biology. Some basic aspects of their reproduction are well-characterized; for example, gestation in deer mice is approximately 23 days, and we know that concurrent lactation and gestation is common in lab-housed stock. We also know that fetal development in deer mice largely parallels house mouse development with a few differences in the timing [13].

However, there are also notable gaps in what we know about fertility and pregnancy in deer mice, and there are no detailed descriptions of placentation in this species. The *P. maniculatus* placenta was first described in detail nearly 60 years ago by Enders [14]. However, since then, only a single detailed study of *Peromyscus* placentas has been published [15], and these descriptions only addressed near-term placentas. As such, there remained a striking gap in comparative understanding of placental development in *Peromyscus*.

A deeper understanding of reproductive biology is fundamentally important for both evolutionary biologists and researchers interested in fitness through time (i.e., [16,17]) as well as disease and community ecologists interested in demographic processes. Reproductive biology (and the ability to predict or control reproduction) is furthermore central to developing genetic tools for the species or performing targeted experimental work in laboratory settings and for understanding reproductive pathologies (e.g., [18,19]). Although the first transgenic embryos for deer mice were produced nearly 15 years ago [20], no transgenic models have followed, despite speculation more than a decade ago that these tools for deer mice were only years away [21,22]. Advancing our comparative understanding of the physiology and development associated with deer mouse reproduction thus has significant potential to advance the utility of the deer mouse as a model across biology.

To address these significant gaps in knowledge, we have generated a gestational atlas for deer mice that (1) assesses the utility of cyclicity as an indicator of fertility, (2) characterizes how body temperature changes across a reproductive bout, and (3) provides a detailed histological description of placental development. We discuss these findings in a comparative context to highlight unique or notable features of deer mouse reproductive biology. In their totality, the data and summaries presented here provide a robust characterization of gestation and development in deer mice that is of-value to both basic researchers studying reproductive ecology or evolution and applied scientists that would benefit from advances in genetic tools and techniques available in deer mice.

## Materials and methods

### Animal breeding and handling

All experimental animals were derived from wild-caught populations trapped in central Nebraska (40°43' N, 99°04' W). Experimental animals were at least three (3) generations removed from wild population and > 90 days old before commencing experimental work. All pregnancies were generated from monogamous pairs (a single female and a single male). Animals were housed at room temperature (23°C) under a 12:12 LD cycle and with *ad libitum* food and waterAnimals were left nearly completely undisturbed during experimental pairing, gestation, and lactation outside of daily welfare checks and cage changes. The wild-derived strain used in this study tends to be highly sensitive to handling; we thus minimized disturbance or handling to avoid potential interference with reproductive parameters including body temperature. Immediately prior to sample collection, animals were humanely euthanized using isoflurane overdose followed by a secondary method (rapid decapitation or cervical dislocation). All animal work was carried out under IACUC-approved protocols at Colorado State University.

Whole implantation sites and maternal tissue and blood samples were collected from dams based on estimated gestational staging (from prior litters, using body temperature patterns, or using visual cues). Gestational timepoint was confirmed by developmental staging of the *in utero* fetuses (following [13,23]). We aimed to collect whole implantation sites from the beginning of placental development (~e11.5 in *Peromyscus maniculatus*) through late term, at which point the placenta is relatively stable in size and appearance. We thus collected sites approximately every 24 h across development. Whole implantation sites were frozen in isopentane on dry ice for immunohistochemistry. All tissues were stored at -80ºC until further processing. Dam and implantation site samples sizes per gestational stage are summarized in S1 Table.

### Body temperature monitoring

To collect continuous measures of body temperature in reproductive females, 14 virgin females received an intrascapular subcutaneous temperature-sensitive PIT-tag (FDX-B PIT tag, Unified Information Devices, Inc.) under light anesthesia with aerosolized isoflurane. Near-continuous body temperature data were collected from animals in home cages using a Unified Information Devices custom-built matrix plate reader system that scanned each cage and recorded temperature approximately every three (3) minutes. Following PIT tag placement, females were singly-housed for at a minimum of five (5) days, after which we added a single adult male (not PIT-tagged). Pairs were left undisturbed except for daily visual checks (cages were not disturbed nor were animals handled) and bi-weekly cage-changes. Five (5) of the 14 PIT-tagged females never became pregnant, as confirmed via dissection after euthanasia. Three (3) females that became pregnant were removed from plates prior to giving birth. The remaining six (6) females that became pregnant remained on plates for temperature collection through birth and weaning of their first litter. Males were left in the cages through birth and lactation. All females became pregnant following the birth of their first litter such that all females were simultaneously lactating and gestating during the experiment. Body temperature recordings were collected through weaning of the first litter (post-natal day 22, pn22), at which time females were euthanized.

We collected an average of 43,776 reads per female (Range: 14,866–60,700 reads) over 55 days (Range: 38–59 days). We experienced one failure of the temperature reading system, which resulted in approximately 24 h of lost data in the middle of a subset of individual experiments (see S1 Fig). Body temperature traces were cleaned of spurious readings using a custom script (available at https://github.com/mjhemmerlein/Peromyscus-Gestational-Atlas). This script retained values for analysis only when they fell within ±1.5×SD of a rolling window-based median (windows size = 11 reads). To avoid artificially high window-based SDs associated with those windows that contained extreme values, the SD used for each individual was determined by calculating the SD for every window and then taking the median value from the full dataset for the cleaning script.

After cleaning, we were left with an average of 40,208 reads from each female (90% of reads retained; Range: 12,083–56,279; 78% - 95%). To visualize and analyze data, we first aligned traces across all females using the estimated day of copulation as an anchor point. Day of copulation was retrospectively estimated using birth date (observed) or developmental stage of *in utero* fetuses while assuming a 23-day gestation period. These timelines follow data from [13,17], which indicate gestation is consistently 23 days in lab-housed deer mice when dams are not concurrently lactating (concurrent lactation and gestation often results in longer gestation periods). For plotting, we then took the minimum temperature in 20-minute bins for each female and averaged this value across females to generate an average minimum temperature for each 20-minute bin across the gestation and lactation period. To determine how daily maximum and minimum temperatures varied across stages of gestation and lactation, we extracted daily minimum and maximum for each female. Differences across stages were evaluated in R 4.0.5 using a linear mixed model (lme4) that included Reproductive Stage (non-pregnant, pre-implantation, gestation, near-birth, early-postnatal, and mid-to-late post-natal) as a predictor and Individual ID as a random effect. Pairwise comparisons were carried out using emmeans() and considered significant at $p < 0.05$.

## Placentation and development

To describe placentation across gestation in deer mice, we generated 10 µm cryosections from whole implantation sites (N ≥ 2 sites per dam per embryonic day) and thaw-mounted sections onto slides. Sections were collected along the approximate midline of each implantation site. We collected at least 3 series of adjacent sections across the mid-line of the implantation site to facilitate multiple labeling protocols. Slides were stored at -80°C until immunohistochemistry.

We employed three separate double-labeling protocols to describe specific features of placentation and development. To examine vascular development and remodeling in the implantation site, we used separate sets of sections to label markers of vascular remodeling and endothelial cell organization: the basement membrane protein laminin and the tight-junction protein ZO-1 [24,25]. To examine interactions between uterine natural killer cells (uNK cells) and trophoblasts in the implantation site, we also labeled one set of sections for the uNK cell marker perforin.

For all immunohistochemistry protocols, sections were fixed with ice-cold 4% paraformaldehyde, permeabilized using methanol, and blocked using 10% normal goat serum in PBS (Vector Laboratories S-1000). Sections were incubated overnight with one of three primary antibodies: rabbit anti-laminin (1:1000, Thermo Scientific PA116730), rabbit anti-ZO-1 (1:500, Invitrogen 617300), or rabbit anti-perforin (1:400, Amsbio TP251). After PBS washes the next day, sections were incubated for 1 hour in a Alexafluor 568 (1:300, goatαrabbit, Thermo Scientific A32731TR). All sections were then labeled for cytokeratin (a trophoblast cell marker) using a 2 hour incubation with pan-cytokeratin antibody conjugated to FITC (1:300, Sigma-Aldrich F3418, lot: 088M4797V) followed by DAPI to visualize nuclei. Immunostained sections were cover slipped with Fluoromount-G, dried overnight at room temperature, and stored at 4°C until imaging on an Evident Scientific APX100. Three sections per implantation site were imaged and quantified for each protocol at one of three objectives (4x, 10x, or 40x). All representative images presented as part of the manuscript are unaltered except for cropping.

Quantification was performed using FIJI (2.14.0/1.54f; Build c89e8500e4) using custom macros, which are available at https://github.com/mjhemmerlein/Peromyscus-Gestational-Atlas. To quantify the relative and absolute growth of placental zones across gestation, we measured the area of each placental zone (decidua, junctional zone, or labyrinth zone) across

three sections per implantation sites and took the median values for each section for plotting. The count and density of uNK cells in the decidua was determined using a watershed method to determine numbers of cells within each image. All zone area and uNK measurements were taken by a single observer.

To quantify the changes in labyrinth zone blood spaces, we used stereology approaches in IMOD version 4.11.25 to determine the percent area of maternal and fetal blood spaces. These procedures were carried out across six 40x images of the labyrinth zone per implantation site. 400 points were used per image and classified as "maternal blood space", "fetal blood space", "tissue", or "nuclei clump". "Tissue" and "nuclei clump" categories were pooled for data analysis and plotting.

Raw data collected from images and used to support qualitative descriptions are available on at https://github.com/mjhemmerlein/Peromyscus-Gestational-Atlas. Raw images from which data were generated are available via Dryad (https://doi.org/10.5061/dryad.z34tmpgr7).

## Results

### Body temperature is a reliable proxy for pregnancy but not fertility in P. maniculatus

Body temperature is a reliable proxy for cyclicity in house mice, and it can also be used to non-invasively detect copulation and successful pregnancies [26]. Others have reported that body temperature may also be used to predict ovulation and successful matings in the closely-related *Peromyscus* species, *P. leucopus* [27]. However, cyclicity was only observed in a subset of individuals [27,28], and it remains unclear as to whether cyclicity is predictive of fertility in *Peromyscus* (i.e., predicts which females are likely to copulate and become pregnant when paired with males).

To answer these questions in deer mice, we collected body temperature traces from 14 virgin female deer mice for five to twelve days (to look for cyclicity) and then paired them with males (to evaluate fertility). We observed overt evidence of cycling from body temperature data in only 4 females (S1 Fig). Addition of the male to the female's home cage significantly disrupted their body temperature rhythms (S1 Fig) – many females shifted their active phase towards their normal rest phase, and their active phase was significantly attenuated.

Nonetheless, we still observed a notable extension in warm body temperature signal beyond the normal offset of activity (~06:00, aligned with lights-on in Fig 1) on the predicted day of copulation among females that became pregnant (Fig 1). We also observed a single incidence of potential pseudo-pregnancy (S1 Fig, sixth panel from the left in the top row). In the instance of apparent pseudo-pregnancy, we observed an extended warm temperature signal similar to that observed on the estimated day of copulation for other females and a relatively warmer rest-phase body temperatures that was initially consistent with patterns we saw in females that became pregnant and later gave birth. However, unlike females that gave birth, the elevated rest phase temperature in this pseudo-pregnant female disappeared within a week. Furthermore, we then observed another extended warm phase aligned with estimated copulation associated with a subsequent successful pregnancy.

Cyclicity was not a useful predictor of whether females were likely or unlikely to produce successful litters): only 2 females that showed body temperature profiles consistent with cycling became pregnant, whereas 7 out of 9 non-cycling females became pregnant after the addition of a male. Based on subsequent birth dates or gestational staging of *in utero* embryos, 8 of these 9 females became pregnant within 3 days of pairing with the male. These patterns were consistent with what we have historically observed in our lab-housed colony: among pairs of deer mice from the same Nebraska population, the median time to first pregnancy was 3 days (Mean±SD: 7 ± 9 days, Range: 0–41 days; N = 49; data deposited online, see **Materials and methods**).

### Daily minimums and maximums vary across physiological stages of gestation and lactation

We next examined daily minimum and maximum temperatures to determine whether there were robust, absolute changes in temperature that could be used to diagnose copulation and gestation progression. Body temperatures were highly variable among individuals and across gestation and implantation; the inter-individual range remained

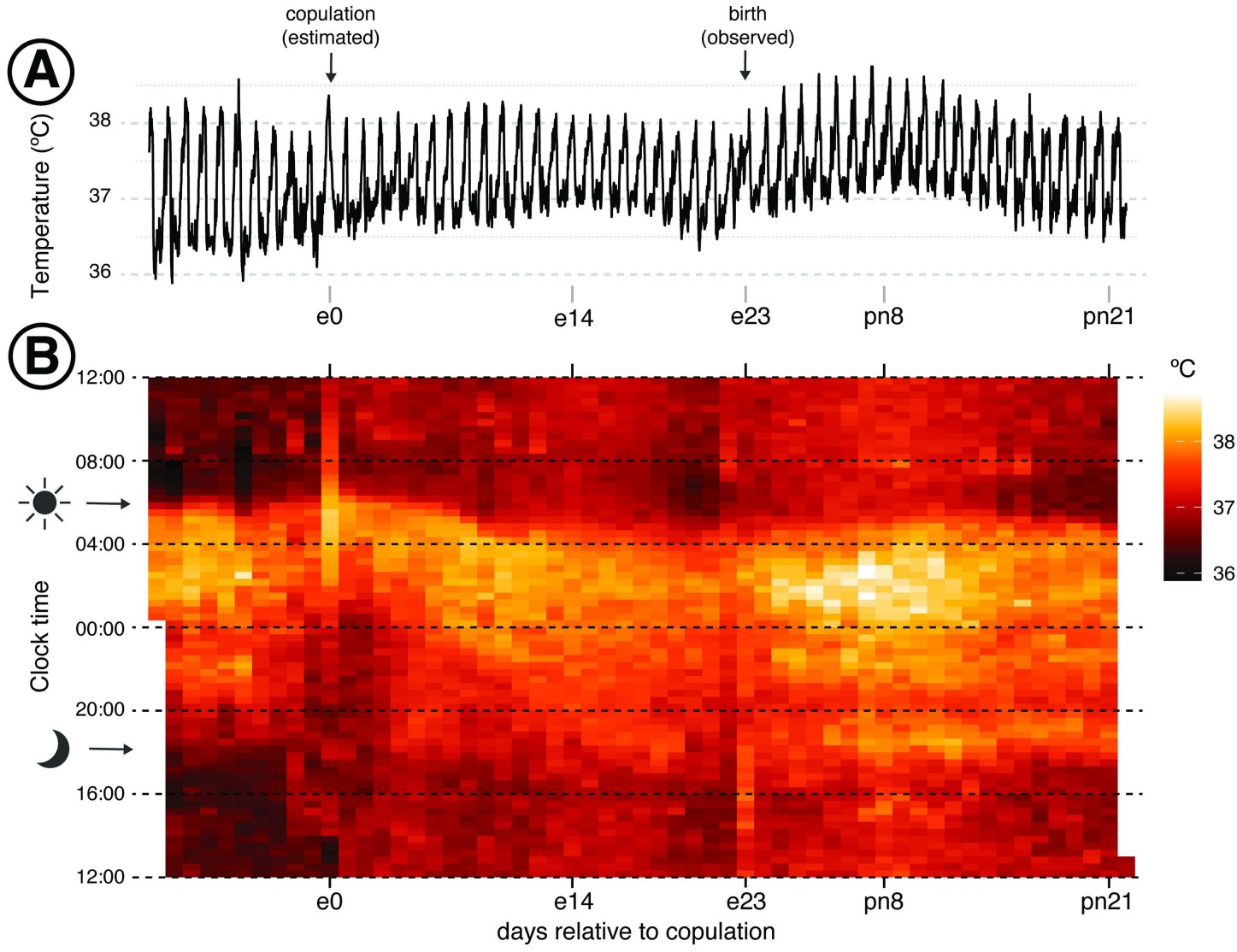

**Fig 1. Body temperature of *P. maniculatus* dams across pregnancy and lactation. (A)** Daily rhythm of body temperature across pregnancy and lactation in deer mouse dams. The line was generated by averaging 20-min. minimum temperatures across 6-9 individuals. Embryonic day 0 (e0) corresponds to the estimated date of copulation and embryonic day 23 (e23) corresponds to the observed day of birth. Pups are weaned on post-natal day 22 (pn22). **(B)** A raster plot of the data shown in Panel A. Each column represents a single day, and each row is a 20 min block. Hours of the day run from 1200 (12:00 PM) at the bottom of the plot to 1159 (11:59 AM) at the top of the plot so that the active phase is centered. The arrowheads on the lefthand side indicate the times of lights-off (moon with arrow) and lights-on (sun with arrow), respectively. Warmer colors represent higher body temperatures, and cooler colors represent lower body temperatures.

around 1.5ºC for both minimum and maximum temperatures across reproductive stages (Fig 2). Nonetheless, we did observe a consistent increase in minimum (rest-phase) temperatures of ∼0.25ºC following copulation in females that became pregnant (Fig 2B). Implantation (conservatively designated here as e6 in deer mice, [22]) resulted in an increase of another 0.25ºC to 0.5ºC above non-pregnant minimum daily temperatures (Fig 2B). This increase was maintained through e20 (Fig 1). Across this same period, maximum daily body temperature declined slightly (∼0.25ºC; Fig 2A). Although the changes in body temperature were relatively small, they were visually noticeable in plots (Fig 1A,1B).

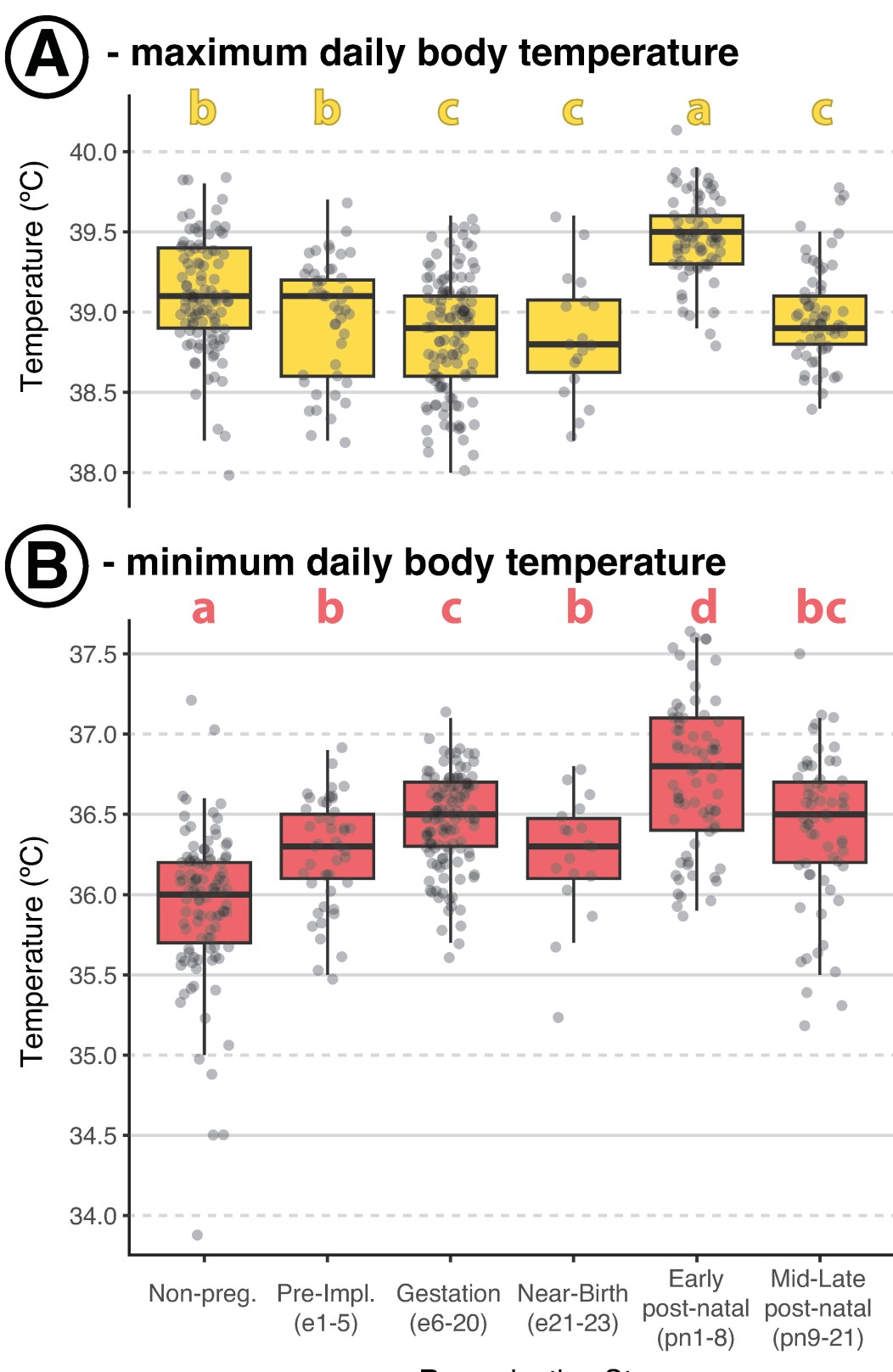

**Fig 2. Daily minimum and maximum body temperatures vary across gestation and lactation in deer mice. (A)** Maximum daily temperatures declined slightly across gestation and peaked during early post-natal development (lactation). **(B)** Minimum daily temperatures increased across the

pre-implantation (Pre-Impl.) and Gestation, but declined Near-Birth. Minimum body temperatures were warmest during the early post-natal period and declined slightly during the mid-to-late post-natal period. The embryonic days (e) relative to copulation or post-natal days (pn) relative to birth are specified below each stage. Different lowercase letters above each boxplot indicate statistically significant differences in daily maximum or minimum body temperature across stages (P < 0.05) in a linear mixed model that included dam ID as a random effect. Shared letters indicate no significant difference between groups. See Materials & Methods for further detail. Each point represents a single day's maximum or minimum temperature per female (N = 6-9 females per phase).

As females approached birth (e21-e23), daily minimum body temperatures declined, and daily maximums remained low (Fig 2). These changes resulted in a clear decrease in daily variation in body temperature (Fig 1A).

The day of birth was marked by an extended warm phase (Fig 1) that continued into the rest phase. Both daily minimum and maximum temperatures were consistently and noticeably elevated thereafter, especially for the first 8 days of lactation; daily minimums were elevated by 0.75ºC and maximums were elevated 0.5 ºC above those observed in the same females prior to pregnancy (Fig 2). Both began to decline 8 days following birth, with maximum daily body temperatures declining by twice as much as minimum body temperatures (0.5ºC vs. 0.25ºC). This transition point at post-natal day 8 (pn8) corresponds to the developmental point at which pups gain directional mobility, including the capacity to leave and return to the nest or seek out siblings independently (*pers. comm.*, M.Y. Juergens). However, their eyes remain closed for another 4 days, and pups do not consume any food beyond the dam's milk until pn15 (*pers. comm.*, M.Y. Juergens).

### Development of the placenta from implantation to near-term

To describe normal placental development in deer mice, we generated a time course of placental development from e11.5 (the developmental stage at which the placenta begins to emerge) through near-term (Fig 3 and 4). In this section, we discuss both gross development as well as specific processes associated with placentation, including vasculogenesis in and around the implantation site and the behavior of uNK cells that regulate implantation in other species (e.g., [29,30]). We examined vascular development and remodeling in the implantation site using two immunohistochemistry protocols that labeled for either laminin (a basement membrane marker) or ZO-1 (a tight-junction protein associated with vascular remodeling and growth [25]). We also examined interactions between uNK cells and trophoblasts in the implantation site using a protocol that labeled sites for perforin (uNK marker) and cytokeratin (trophoblast marker).

We also describe the appearance and behavior of trophoblast subtypes across placentation in the deer mouse. Trophoblast cells derived from the blastocyst differentiate into a variety of sub-types that carry out specific roles in establishing the implantation site, mediating maternal-fetal interactions at the cellular interface, placental development, and vascular remodeling [31–34]. We identified four distinct trophoblast giant cell (often abbreviated TGC) types (primary, migratory, parietal, and canal) as well as some invasive or migratory trophoblast cell types and the syncytiotrophoblasts lining the exchange surfaces in the labyrinth zone [35]. Throughout, we describe some notable ways in which trophoblast behavior in the developing deer mouse placenta differ from patterns observed in house mice and other cricetids. We summarize major placentation events and their comparative timing between deer mice and house mice in **Table 1**.

### Early development (e11.5, e12.5)

Implantation sites exhibited clear polarity in the decidua surrounding the trophectoderm by e11.5 – the decidua developing on the mesometrial side of the blastocyst was at least 2 times thicker than at other contact points. The decidua contained dense and expansive endometrial glands that are likely producing histotroph for the developing embryo (Fig 3, e11.5). These glands were most abundant at e12.5, with up to 60% of the decidua proper containing glands.

Similar to other rodents [36], deer mouse trophectoderm on the mesometrial side of the implantation site was organized into a proliferative and invasive ectoplacental cone comprised of trophoblast cells early in development (Fig 3). Primary trophoblast giant cells were the first to differentiate from the progenitor trophectoderm. These giant cells appeared within

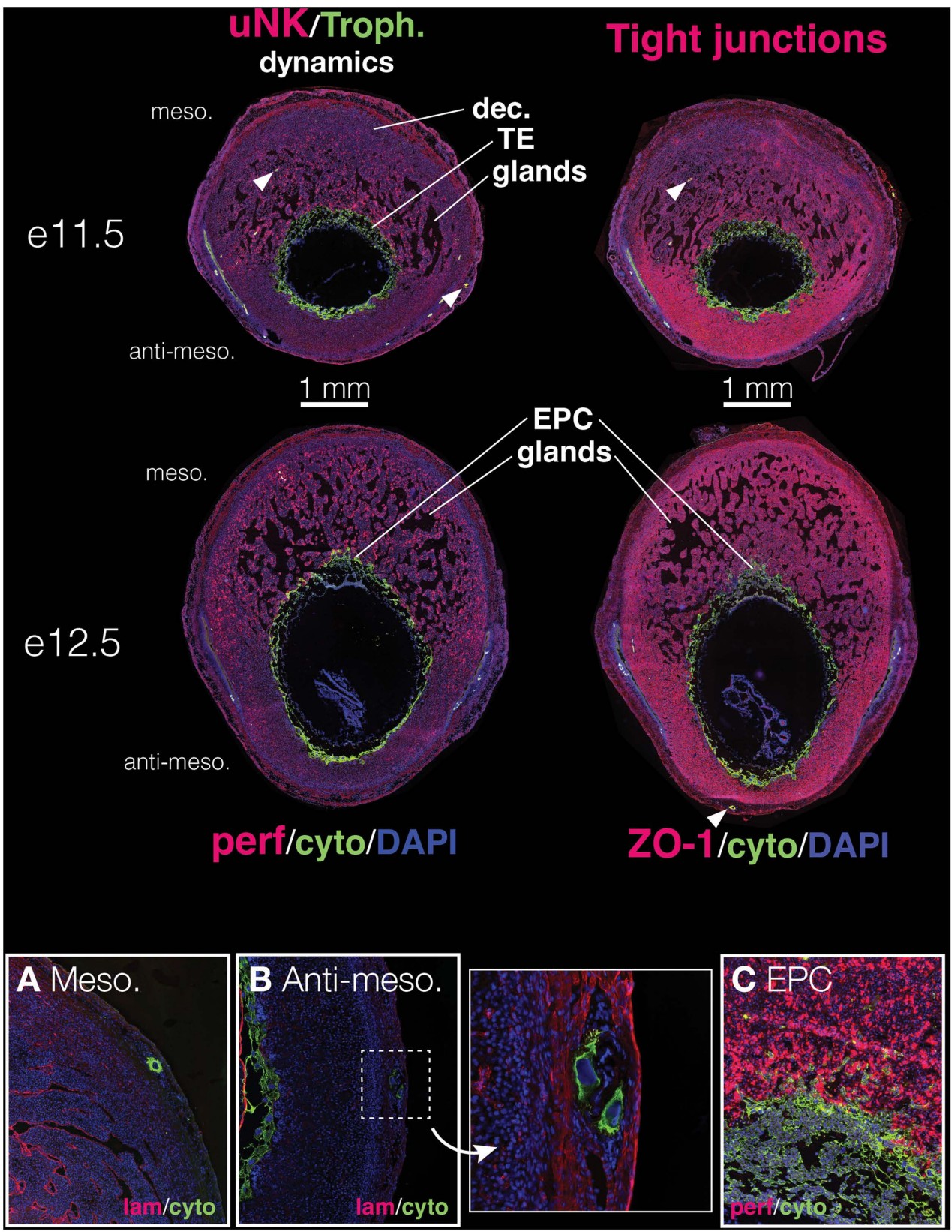

**Fig 3. Representative immunohistochemistry of the deer mouse placenta in early development (embryonic days 11.5, 12.5).** Representative images show both the general developmental patterning **(Top)** and specific behavior of invasive giant trophoblast cells **(Bottom, A,B)** and uterine natural killer cells (uNKs; **Bottom, C**) during this period. The specific proteins that were labeled in each series are indicated below each set of figures. See Materials and Methods for more. **(Top)** During early placentation and development, only the decidua **(dec.)** and trophectoderm **(TE)** are apparent; the zones of the fetal placenta have not yet developed. Throughout these early stages, the decidua contains large and extensive endometrial glands **(glands)** that provide early nutrition to the developing embryo. **White arrowheads** indicate large, cytokeratin-positive invasive giant trophoblast cells. Scale bars (1 mm) are shown immediately below e11.5 tissue sections. **(Bottom, A-C)** Higher magnification images show detail of invasive giant trophoblast cells and uNKs. Invasive giant trophoblast cells, which are distinguishable by their large nuclei and cytokeratin-positive cell body, occur both in the mesometrial side of the implantation site **(A, Meso.)** and the anti-mesometrial side **(B, Anti-meso.)**. During this early period of establishing the implantation site, uNKs exhibit intimate interactions with the ectoplacental cone **(EPC)** of the trophectoderm **(C)**.

the maternal decidua and even as deep as the maternal myometrium by e11.5 (Fig 3A). Primary trophoblast giant cells also appeared within the anti-mesometrial side of the implantation site up through e12.5 (Fig 3B), likely arising in these areas as part of early stages of implantation and decidualization initiation [36].

ZO-1 displayed marked polarity in expression across the implantation site (Fig 3). The mesometrial decidua dsiplayed diffuse ZO-1 expression with considerable spatial variation in ZO-1 intensity among cells surrounding the glands. This pattern is consistent with the dynamic regulation of permeability along these glands during this period. In contrast, on the anti-mesometrial side, ZO-1 was extremely densely expressed, suggesting there are robust and pervasive tight junctions throughout the decidual cells there. Nonetheless, we still observed giant trophoblasts within this area and into the myometrium on the anti-mesometrial side of the implantation site, suggesting that trophoblast giant cells were still able to migrate through this tissue. Indeed, the trophectoderm on the anti-mesometrial side of the implantation site generally had a much more irregular border relative to the mesometrial surface. Moreover, we observed similar numbers of migratory trophoblast giant cells on both the mesometrial and anti-mesometrial sides. The tight junctions formed via ZO-1 thus do not seem to preclude migration of trophoblast giant cells through this tissue.

On the other hand, uNKs showed strong polarity in abundance across the implantation site that was inverse to ZO-1 expression on e11.5 (Fig 3). uNKs were extremely abundant within the mesometrial decidua by e11.5 but largely absent from the anti-mesometrial decidua, where ZO-1 was most abundant. uNKs did not exhibit any noticeable organization or patterning within the mesometrial decidua, though they may have been more closely clumped nearer to the trophectoderm-decidua interface. At this timepoint, we did not observe any non-giant cell invasive trophoblasts.

By e12.5, the ectoplacental cone had expanded along the trophectoderm/decidual interface on the mesometrial side, and trophoblasts had emerged from the ectoplacental cone and migrated into the decidua (Fig 3C). The anti-mesometrial trophectoderm was still comprised of primary trophoblast giant cells, however trophoblasts with small nuclei emerged within the ectoplacental cone on the mesometrial side at e12.5 (Fig 3C). Tight junctions appeared more prevalent throughout the decidua by e12.5; their expression on the anti-mesometrial side remained extremely high and dense (Fig 3), but they were increasingly abundant on the mesometrial decidua as well, especially in the periphery (i.e., further away from the ectoplacental cone). Conversely, uNKs continued to expand in abundance and distribution between e11.5 and e12.5, surrounding the implantation site at e12.5, even though they remained much more diffuse on the anti-mesometrial side. They were also still largely absent from the immediate vicinity of the embryo on the anti-mesometrial side. On the mesometrial side, they appeared to be in direct contact with the ectoplacental cone (Fig 3C).

### Emergence and maturation of the placenta (e13.5 – 17.5)

The definitive zones of the placenta (namely, the decidua, junctional zone, and labyrinth zone) began to differentiate from the early trophectoderm on e13.5 of gestation (Fig 4). On e13.5, we observed a distinctive band of trophoblast giant cells adjacent to the decidua that subsequently expanded and differentiated to form the junctional zone (Fig 5A,5B). The junctional zone continued to expand in size until around e16.5, after which it was stable or declined in size (Fig 6, Top).

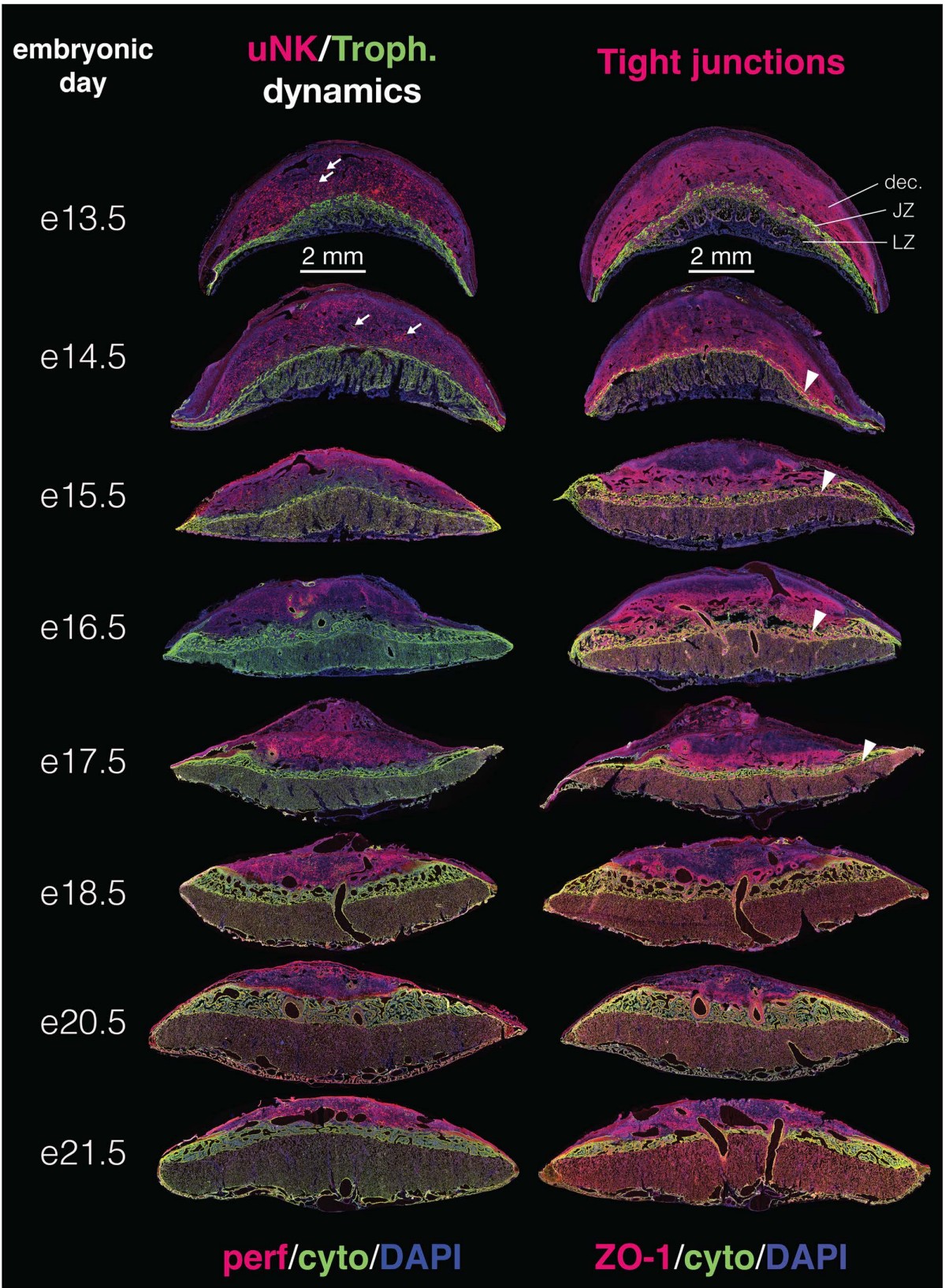

**Fig 4. Representative immunohistochemistry of the deer mouse placenta from embryonic day 13.5 (e13.5) to e21.5.** Representative images show general patterning and developmental changes in interactions between uterine natural killer cells (**uNKs**) and trophoblasts (**Troph.**) (**Left**) and tight

junctions (**Right**) across placental compartments. The specific proteins that were labeled in each series are indicated below each set of figures. See Materials and Methods for more. Scale bars (2 mm) for all sections are provided immediately below e13.5 tissue sections. Major zones of the placenta are indicated in the top right (decidua, **dec.**; junctional zone, **JZ**; labyrinth zone, **LZ**). **White arrows** in the left series of images point to early emergence of maternal vasculature within the decidua that are lined by trophoblasts. **White arrowheads** in the right series of images point to ZO-1-positive cell clusters within the developing junctional zone. See main text for further discussion. For a higher resolution version of this image, see **Supporting Information**.

**Table 1. Comparative timing of key developmental events in placentation between house mice (*Mus musculus*) and the North American deer mouse (*Peromyscus maniculatus*).**

| Event | *Mus musculus* (from [36] | | *Peromyscus maniculatus* | |
|---|---|---|---|---|
| | Embryonic day (from copulation) | Fetal development (Theiler Stage) | Embryonic day (from copulation) | Fetal development (Theiler Stage) |
| Primary and parietal trophoblast giant cells s apparent | 5.5–6.5 | 8 | 13.5 | 18 - 19 |
| Induction of the labyrinth zone | 8.5–9.5 | 11-13 | 13.5 | 18 - 19 |
| Junctional zone development | 8.5–9.5 | 11-13 | 13.5–14.5 | 19 - 20 |
| Maternal spiral arteries developing in decidua | 10 - 12 | 16-20 | 13.5 | 19 |
| Maternal spiral arteries lined by trophoblasts (decidua) | 11.5 | 18-20 | 16.5 | 20-21 |
| NK cell abundance plateaus | 12.5-13.5 | 20-21 | 17.5 | 22 |
| Labyrinth zone peaks in size | 14.5 | 22 | 18.5 | 23 |
| NK cell abundance begins to fall | 15.5–17.5 | 23-26 | NA (remains high through term) | |
| Decidual tissue declines | 15.5–17.5 | 23-26 | 18.5–21.5 | 23-26 |

We observed striking expression of ZO-1 within the developing junctional zone (arrowheads in Fig 4). These ZO-1-positive cells displayed distinct organization into clumps starting around e14.5, but they disappeared from the junctional zone just after e17.5. ZO-1 may be a marker for trophoblast differentiation [37], and thus the loss of ZO-1 could reflect the maturation of the junctional zone.

The placenta as a whole grew dramatically in size through e16.5 as well (Fig 6, Top), though most of this growth was associated with development and expansion in the labyrinth zone. The labyrinth zone increased nearly 10 fold in size between e13.5 to e17.5 (Fig 6, Top). Initially (e13.5, e14.5), the labyrinth was predominantly comprised of cytokeratin-positive trophoblasts. The early fetal vasculature emerged as involutions along the chorionic plate that were lined externally by trophoblasts and internally by laminin-positive fetal endothelial cells (Fig 5B). These structures began to undergo branching morphogenesis to create the labyrinthine structure that will be perfused by maternal blood later in the gestation. The branching morphogenesis process appeared to be extremely rapid, and the majority of the labyrinth zone was filled by these complex structures by e14.5 (Fig 4). However, the progenitor vascular structures also expanded and persisted within the labyrinth zone well-beyond this early period, through at least mid-gestation (e16.5-e17.5; Fig 4 and 5C). At first, these early fetal blood spaces were filled with nucleated red blood cells (Fig 7). Nucleated cells remained common within fetal vascular spaces through e17.5, however they decreased in abundance (i.e., became less numerous) after this point and were not readily observed within the tortuous fetal blood spaces thereafter.

Up through e16.5, the majority of LZ expansion was associated with the branching morphogenesis and expansion of fetal blood spaces alone (Fig 7). Maternal blood spaces within the labyrinth zone only began to expand substantially after e16.5, and they exhibited a dramatic expansion after e17.5 (Fig 7). The notable expansion after e17.5 was aligned with several other developmental events discussed in the following paragraphs that all suggest that, around e17.5, the placenta and fetus transition to hemotrophic nutrition through the mature labyrinth zone. We therefore predict that the

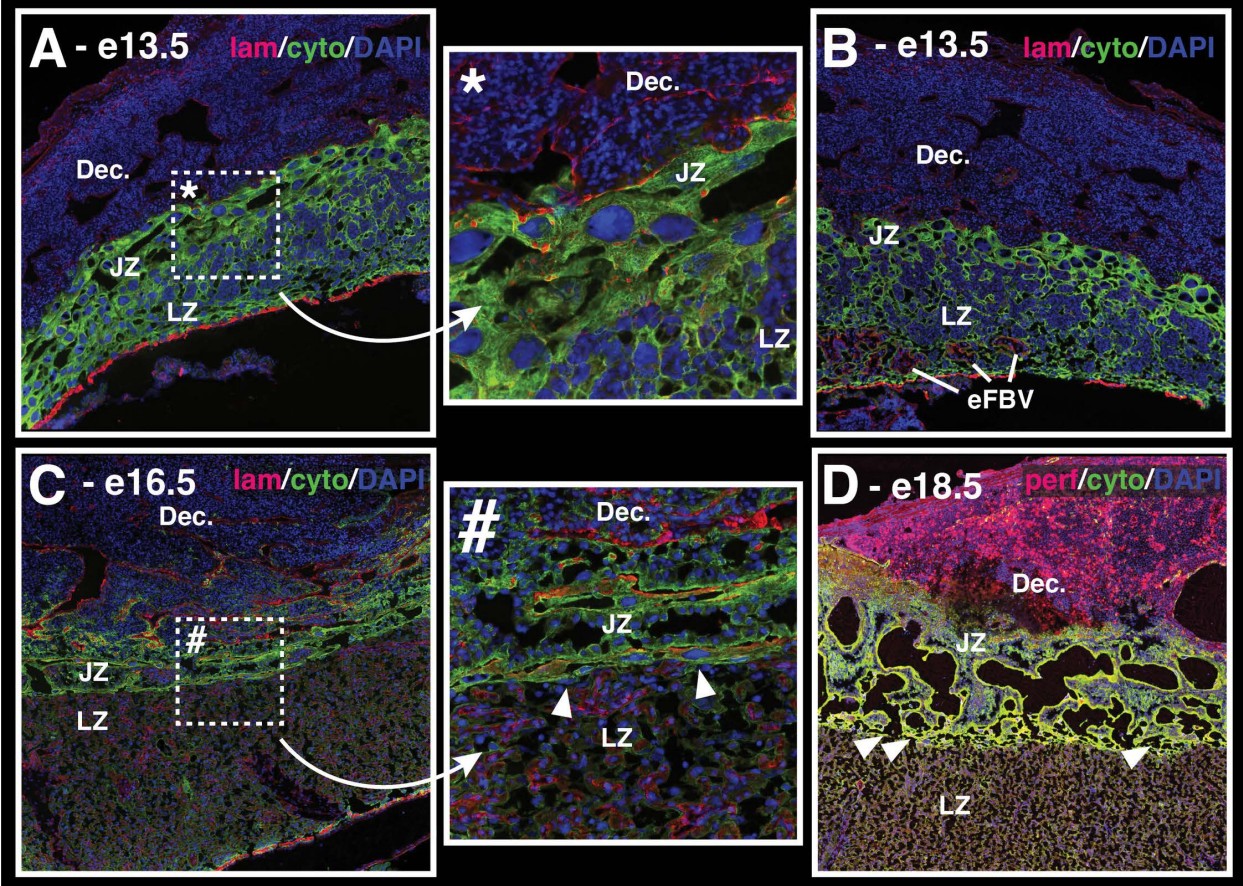

**Fig 5. Development of the junctional and labyrinth zones in the *P. maniculatus* placenta. (A)** The early junctional zone (JZ) is comprised of cytokeratin-positive trophoblast giant cells (TGCs) while the early labyrinth zone (LZ) contains a mix of TGCs and smaller trophoblast types. **(* inset)** shows detail of JZ TGC structure. **(B)** LZ development continues with emergence of early fetal blood vasculature (eFBV) from the chorionic plate that expand as development continues. **(C)** As the LZ matures, a barrier wall of JZ TGCs forms along the LZ and JZ interface (arrow heads in **# inset**), while the interface between the JZ and decidua (Dec.) remains somewhat diffuse. **(D)** By e18.5, the TGC barrier between the JZ and LZ begins to degrade to create space for blood flow between the maternal vascular lacunae in the JZ and the mature LZ (arrow heads), suggesting an onset of significant hemotrophic nutrition. The placental sections in **Panels A-C** are labeled for laminin (red, fetal endothelial cells), cytokeratin (green, fetal trophoblasts), and cell nuclei (blue, DAPI). The placenta shown in **Panel D** is labeled for perforin (red, uterine Natural Killer cells), cytokeratin (green, fetal trophoblasts), and cell nuclei (blue, DAPI).

increase in maternal blood space volume was, in part, associated with a dramatic increase in maternal blood volume and flow through the labyrinth zone beginning around e17.5.

First, maternal vasculature within the implantation site and decidua were fully established by e17.5 to complete a fully-functional maternal vascular circuit. Maternal arteries first became apparent in the decidua around e13.5, and we observed near-immediate associations between these vascular structures and trophoblasts (cytokeratin-positive cells) (arrows in Fig 4). Some of these spaces also showed a loss of laminin expression where basal lamina would be (Fig 7), suggesting that trophoblasts weere replacing endothelial cells that lined maternal vasculature. On e13.5 and 14.5, there were very few invasive trophoblasts in the decidual stroma – we only observed trophoblasts associated with vasculature at this stage. The decidual stroma remained dense with uNKs during this period, though we did not observe distinct associations between uNKs and maternal vasculature or invasive trophoblasts, as might be expected if uNK-trophoblast interactions played a role in vascular remodeling in deer mice.

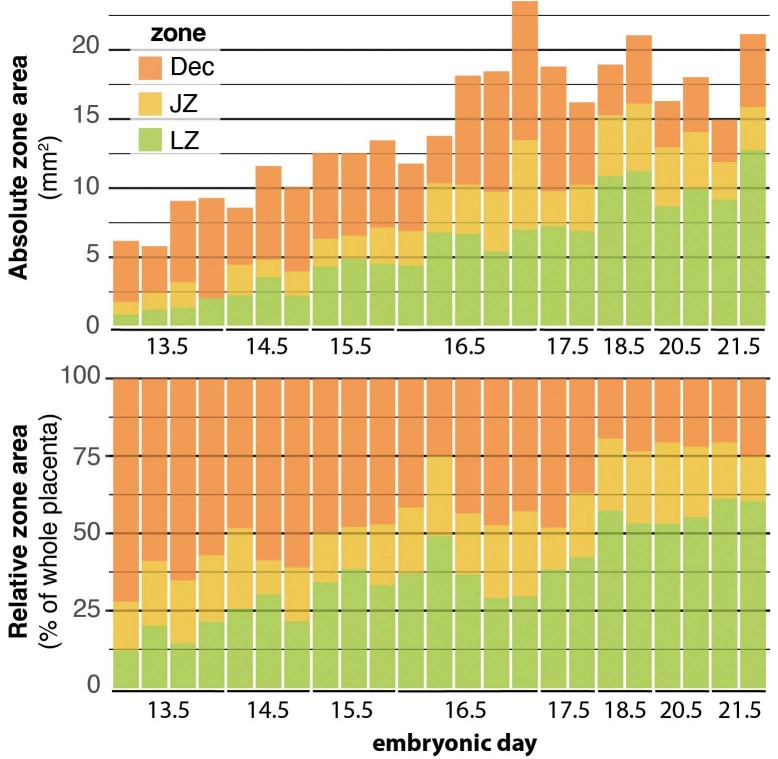

**Fig 6. Changes in the absolute and relative size of placental zones across gestation.** Each vertical bar shows average zone sizes for a single implantation site. **(Top)** Averages of the absolute zone area was taken from measurements collected across 3-6 sections from each implantation site. **(Bottom)** The relative zone area was calculated as average zone size relative to the whole placental area (the sum of the zones shown in the top figure).

By e15.5, the abundance of invasive non-giant cell trophoblasts in the decidual stroma increased dramatically. This increase occured in parallel to a retreat of uNK cells – although uNKs still appeared along the junctional zone-decidua interface, they were much sparser. We also observed an increase in the number of vascular spaces in the decidua and in the number of those that were trophoblast-lined and laminin-negative at e15.5. Vasculature that were lined by trophoblasts still retained strong expression of ZO-1.

On e16.5, the maternal vasculature in the decidua began to undergo noticeable increases in diameter associated with trophoblast remodeling, and the major maternal canal that delivers blood from the labyrinth zone to maternal circulation [23] was established by e16.5 in all implantation sites. The maternal canal was surrounded by a combination of trophoblast giant cells and syncytiotrophoblasts, though the syncytiotrophoblasts were more abundant. The canal was further surrounded by several layers of cells that were initially strongly-positive for ZO-1 (i.e., tight junctions) and laminin-negative (there are no endothelial cells lining the canal). The diameter of the maternal vasculature in the decidua and the maternal canal increased in diameter through e17.5, and just after e17.5, all of these vessels were predominantly lined by trophoblast cells.

Second, trophoblast development and behavior within the junctional zone across this period also point to e17.5 as the transition point from histotrophic to hemotrophic nutrition, however there appeared to be some unique aspects of this process in deer mice. In most rodents, parietal trophoblast giant cells form the barrier between the junctional zone and decidua [23], however, this organized layer did not appear in the deer mouse placenta. Instead, we observed an accumulation of parietal trophoblast giant cells at the edges of the placental disc in deer mice around e14.5, and e15.5 (S2 Fig); these accumulations of parietal trophoblast giant cell have also been observed in other cricetids [38], and they may be

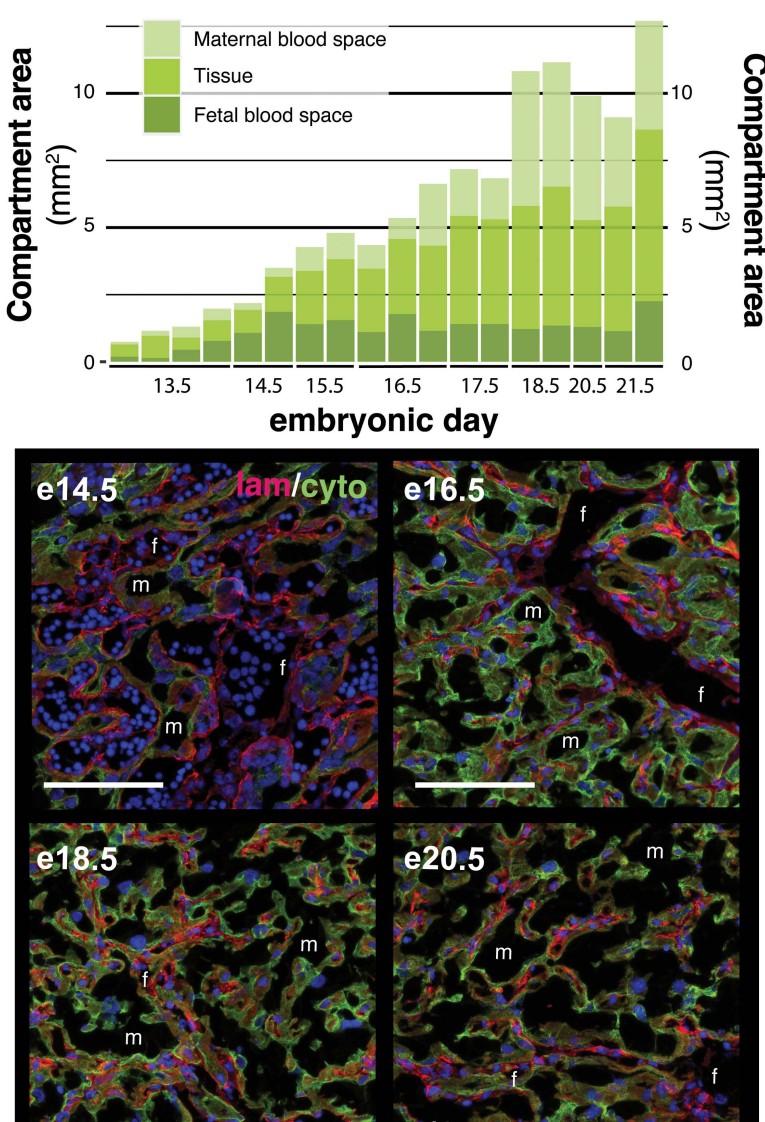

**Fig 7. Development of maternal and fetal blood spaces in the *P. maniculatus* labyrinth zone. (Top)** Quantitative changes in the total area occupied by maternal blood space, tissue, or fetal blood spaces in the labyrinth zone across gestation. Each vertical bar shows average zone sizes for a single implantation site. The total compartment area was calculated by multiplying the percentage area from stereology measurements taken at high magnification (40x; see Methods) by the total zone size, calculated from the implantation site section imaged at low magnification (4x). **(Bottom)** The early labyrinth zone (**e14.5**) is predominantly large fetal blood spaces (f) that are lined by endothelial cells (laminin-positive) and filled with nucleated red blood cells. These large blood spaces persist through **e16.5**. By e16.5, the maternal blood spaces (m) that are lined by trophoblasts (cytokeratin-positive) are increasing in size and relative area within the labyrinth zone. However, some large fetal blood spaces remain. For the remainder of development, maternal blood spaces appear to expand in a size white fetal blood spaces remain relatively static in area (**e18.5, e20.5**). All placental sections are labeled for laminin (red, fetal endothelial cells), cytokeratin (green, fetal trophoblasts), and cell nuclei (blue, DAPI). As indicated above, representative maternal and fetal blood spaces are labeled in each image using **m** and **f**, respectively.

consistent with thickening of the edges of the parietal trophoblast giant cell layer observed in mice. However, unlike other cricetids or mice, these accumulated trophoblast giant cells did not persist through placental development; the placental disc parietal trophoblast giant cells disappeared by e16.5 in deer mice. We did however observe what appeared to be

parietal trophoblast giant cells forming a distinctive barrier between the junctional zone and the labyrinth zone (Fig 5C). Around e17.5, we observed gaps forming between trophoblast giant cells in this barrier, which created links between blood space in the labyrinth zone and the junctional zone for return of maternal blood to the maternal circulatory system (Fig 5D). We are not aware of other rodents where such a clear trophoblast giant cell-mediated barrier is created and then degraded to control this flow along the junctional and labyrinth interface, but again, this timing is consistent with a completion of the maternal vascular circuit within the implantation site around e17.5.

The third and final developmental event that is consistent with a switch from histotrophic to hemotrophic nutrition around e17.5 is the regression of endometrial glands in the decidua. The endometrial glands that dominated the decidua during early development gradually became less abundant or collapsed from e13.5 to 16.5, such that they were nearly completely absent from the decidua by e16.5. Decidual erosion also occurred across this period, and both processes contributed to the apparent stasis of whole placental size after e16.5, even as the labyrinth zone continued to grow.

Unlike in house mice [36], uNK cells remained abundant and broadly distributed in the decidual stroma throughout this process. uNK numbers and density peaked around e16.5, however we did not observe any noticeable decrease in their density thereafter (Figs 4 and 5).

### Near-term changes in the placenta (e18.5 – e21.5)

On e18.5, we observed enlargement of large maternal blood spaces within the junctional zone that appear to be completely laminin-negative, likely in-line with the active maternal vascular circuit through these spaces.

Major maternal vascular structures did exhibit some thinning along their walls, particularly along the maternal canal, as gestation approaches term. In addition, tight junctions (ZO-1) along this vessel in mid-gestation (e16.5) were largely absent within the labyrinth zone proper in late gestation, and their intensity was much reduced within these surrounding cell layers through the junctional zone and decidua. Nonetheless, a robust "band" of ZO-1 positive stromal cells in the decidua remained present along the junctional zone-decidua interface, and the intensity was particularly notable around maternal vascular spaces therein.

We observed that the absolute size of the developing and mature placenta was quite variable across implantation sites, however the *proportion* of the placenta that was allocated to each compartment was similar, especially by e18.5 (Fig 6, Bottom). Overall, the mature placenta remained fairly stable in size after e18.5 – the placenta did not appear to decline or deteriorate in the final gestational period (Fig 6, Top).

## Discussion

The North American deer mouse shares many useful reproductive features with other common lab models, including the house mouse. As is true in house mice [26], body temperature appears to be a useful marker of successful copulation in the deer mouse, and body temperature (especially daily minimums) may be used to track pregnancy progression in this species. We also found many fundamental similarities in placentation and the organization of the implantation site across development relative to house mice. However, North American deer mice also display some unique traits in both reproductive cyclicity and placental development that are important for researchers interested in reproductive physiology as well as those interested in developing new reproductive technologies for the species.

### Core body temperature across gestation

In many mammals, core body temperature is a reliable read-out for female fertility because hormonal changes following ovulation promote an increase in basal body temperature (temperature during the rest-phase). When individuals become pregnant following ovulation, body temperature remains high because these hormonal signals persist (i.e., an individual becomes pregnant or pseudo-pregnant) [26,27,39–41]. In contrast, for species that spontaneously ovulate, pregnancy

loss or failure for egg and sperm to fuse results in a loss of the hormonal signals that produce elevated body temperatures, and thus body temperature falls. As such, for humans and strains of house mice that have spontaneous ovulation, females display cyclical patterns in body temperature as they cycle through stages of egg maturation and ovulation.

Our temperature data showed some evidence for spontaneous ovulation (i.e., repeated estrous cycles) in deer mice (see S1 Fig), but the majority of the females in our study did not show any sign of spontaneous ovulation. Moreover, spontaneous ovulation did not predict fertility – most of the females that showed no sign of spontaneous ovulation in our study still became pregnant, and many were receptive to copulation within hours of the male being added to the cage. These results align with prior reports suggesting that only some females among *Peromyscus maniculatus* and the closely-related *P. leucopus* spontaneously ovulate [21,22,27,42]. However, some of these prior studies have assumed that the absence of spontaneous ovulation indicated lower reproductive readiness or receptivity. Instead, our data suggest that deer mice can and may more commonly exhibit afacultative or opportunistic reproductive strategy. This variation could indicate differences in sensitivity to male presence or environmental cues that regulate ovulation and reproductive physiology more generally (as suggested by [42]).

These are at least two other potential explanations for the observed patterns in body temperature and reproductive success. First, ovulation may always be spontaneous in deer mice, but the subsequent hormonal changes that drive whole-animal changes in temperature and activity may depend on additional cues, like male presence. Such a mechanism would depend on action at or in the corpora lutea, the endocrine structure derived from the ovulated follicle that is responsible for producing the progestogens that lead to altered body temperature patterns [43]. Second, the formation of corpora lutea (which has been cited as independent evidence of spontaneous ovulation, [42]) may not be ovulation-dependent. We already know that formation of these endocrine structures can arise from non-ovulatory follicles in other mammals [43,44]. Although these non-ovulatory corpora lutea are usually observed in the context of pregnancy [43,44], it may be incorrect to assume that these structures can only form in that context. Non-ovulatory corpora lutea formation in deer mice could also help explain the fact that oocytes are recovered at very low rates or even not at all from deer mice, even when researchers have observed ovarian corpora lutea [42]. Further work on the ovarian endocrinology and regulation of female reproduction in deer mice will be necessary to differentiate among these physiological mechanisms. Such comparative research may also help expand our understanding of variation in reproductive strategies among mammals.

We also observed notable changes in body temperature across gestation and lactation, including both maximum and minimum body temperatures. The reduction in maximum body temperature near-birth has been noted in other mammals [41,45]. The mechanisms for this decrease are not clear, though it may be linked to hormonal shifts that precede and ultimately induce parturition. Reductions in maximum body temperature could also be associated with reductions in activity; deer mice often gain ~50% of their pre-pregnancy body weight by the time of birth, and thus their mobility is significantly limited. Indeed, lower body temperature in late gestation coincides with decreased activity in house mice [46]. We did not record activity, so we cannot determine whether activity alone could account for these differences. Definitive resolution of the drivers of these striking changes in body temperature across gestation and lactation require follow-up efforts to quantify behavior, activity, and metabolism. Field studies would also be beneficial for determining whether the reduction in temperature observed in the lab corresponds to patterns in free-living animals which must continue to forage for food.

Finally, we observed a strong peak in both rest and active-phase body temperature associated with the early prenatal lactation period. These changes are predicted to be associated with the dramatic metabolic demands of lactation. However, we found that both maximum and minimum temperatures began to dissipate only 8 days after birth (pn8). Deer mouse pups do not begin to feed themselves from the hopper until pn15, and thus we know dams are providing all of the nutrition for their rapidly growing pups through at least pn14 (*pers. comm.*, M.Y. Juergens). Our temperature data thus suggest the strong increase in body temperature observed in the first 8 days is a function of factors other than simply the metabolic demands of producing and secreting milk. Such factors could include mammary gland growth and maturation, which may be complete by pn8, a shift in metabolic strategies from a mobilization of internal lipid stores (i.e.,

capital-based) to processing food in real time (i.e., income-based), or even a shift in the extent to which pups rely on maternal thermogenesis for thermoregulation.

## Placentation and development

Our histological atlas suggests that, in general, placentation and development in the North American deer mouse is grossly similar to placentation in other *Muroid* rodents (Table 1) [36,47,48]. The deer mouse placenta also has some traits that are apparently unique to Cricetids, including the accumulation of parietal trophoblast giant cells on the edge of the placental disc [38,49] and the persistence of large, progenitor fetal vasculature in the labyrinth zone late into development. Although not explicitly discussed by the authors, there are grossly similar fetal vascular spaces in the South American Sigmodont *Necromys lasiurus* (the Hairy-tailed akodont) in mid-gestation [38]. The extent to which these Cricetid-specific features are functionally important (i.e., endocrinologically active or structurally functional) would be of value to understanding the evolution of placentation across mammals.

We also observed a few aspects of placentation that may be unique to deer mice or their genus (*Peromyscus*). First, we observed more trophoblast-directed vascular remodeling in the deer mouse than has been described in other Cricetids. In South American cricetids, decidual arteries are largely thought to be remodeled by uNK cells [38], however we did not observe similar associations between uNK cells and decidual vasculature in the North American deer mouse. These cell type-specific roles are again important for understanding comparative placentation writ large, and for choosing appropriate rodent models to study gestational disease. Further physiological and histological studies are needed to clarify the roles of trophoblasts and uNK cells in vascular remodeling in the deer mouse.

Second, we observed a distinctive layer of trophoblast giant cells between the junctional zone and labyrinth zone boundary immediately prior to the switch to hemotrophic nutrition, which has not been described in any rodent to our knowledge. Moreover, the *absence* of the trophoblast giant cell layer between the junctional zone and decidua starkly contrasts with patterns in Muroids and other Cricetids [38]. These changes may indicate some fundamental differences in the maternal-fetal interface and interactions therein that could be highly relevant to the utility of different rodent models in studying reproductive biology.

Third, we found that the tight junction protein ZO-1 displayed interesting and dynamic expression across gestation in both the decidua and junctional zone. In the decidua, these dynamic expression patterns suggest that tight junctions in the decidua are carefully regulated throughout placental and fetal development. Further comparative work is necessary to understand how generalized these patterns are and their functional relevance. In the junctional zone, the trophoblast-specific ZO-1 expression we observed would be consistent with organization and differentiation of specific trophoblast sub-types, as has been suggested in human placentas [37]. Detailed histology and cell type-specific phenotyping will be necessary to contextualize the universality (or lack thereof) in these trophoblast development dynamics across mammals.

## Limitations of the current study

One important limitation of the descriptions and data presented here is that our data are derived from central Nebraskan deer mice that were maintained in a captive colony for < 5 generations. Deer mice are a highly diverse species across their range, and thus there is good reason to suspect that some of these patterns (especially related to rates of spontaneous ovulation) will vary across populations. We would also expect some of these traits to differ with the stock center deer mouse lines (BW, SM2), which have almost certainly experienced inadvertent selection on some reproductive traits associated with long-term captivity and lab-based breeding [17]. Nonetheless, the patterns we describe here lay a robust foundation for future comparative work across populations that can facilitate advances in understanding the genetic and ecological factors that influence reproductive traits like rates of spontaneous versus induced ovulation.

## Conclusions

The vast majority of what we know about reproductive biology in mammals comes from some primates, common laboratory rodent species, and domestic species. Building a deep understanding of comparative reproductive physiology and how these traits influence evolutionary trajectories depends on expanding beyond these core species to incorporate the breadth of diversity across mammals. For example, understanding the physiological and behavioral contributors to the changes in female body temperature across reproduction may provide novel insight into dynamic investment strategies, limits to sustained energy intake, and individuals' capacity to lactate and gestate simultaneously. Comparative work addressing these patterns will also be important for understanding how and when metabolism constrains mammalian reproduction, and how some species overcome these limits. The North American deer mouse is well-positioned to contribute to a wide range of fields by virtue of its inherent diversity across its broad range. Our findings provide a foundation for supporting both functional and comparative research to build out the deer mouse as a model rodent in the coming decades.

## Supporting information

**S1 Table. Sample sizes for histological data collection across gestational timepoints and dams.**
(DOCX)

**S1 Fig. Raster plots for all experimental females with temperature-sensitive PIT-tags.** Warmer colors represent higher body temperatures and cooler colors represent lower body temperatures. Each column represents a single day, and each row is a single hour. Hours of the day run from 1200 (12:00 PM) at the bottom of the plot to 1159 (11:59 AM) at the top of the plot so that the active phase is centered. Individual temperature raster plots are grouped by type (cycling, not cycling) and outcome (successful matings that resulted in pregnancies or unsuccessful matings). Cyclicity was determined using qualitative assessments of fluctuations in rest-phase body temperature. Estimated days of estrous used to assign cyclicity is indicated using asterisks (*) in plot. All plots use an identical range of temperatures for color plotting (Temp Key in bottom right). White boxes in plots are missing data; we experienced one failure of the temperature reading system, which resulted in approximately 24 h of lost data in the middle of a subset of the experiments.
(TIF)

**S2 Fig. High resolution version of Figure 4.** This file is a larger and higher resolution version of Figure 4 that captures additional detail in the implantation sites shown.
(PNG)

**S3 Fig. Parietal trophoblast giant cells (pTGCs) accumulate at the edges of the placental disc in mid-gestation.** (**Top**) pTGCs accumulate at the edges of the placental disk in the junctional zone on embryonic day 15.5 (e15.5). The expanded image (right) shows further detail. (**Bottom**) Representative images of pTGC accumulation across sections show that the accumulation of pTGCs is common and extends to e16.5. These accumulations are largely ZO-1-negative (bottom right).
(TIF)

## Acknowledgments

The authors thank Chloe E. Butler for her help in developing the script to clean continuous temperature data and Makenna Y. Juergens for her input on post-natal development in deer mice.

## Author contributions

**Conceptualization:** Kathryn Wilsterman.

**Data curation:** Megan J Hemmerlein.

**Formal analysis:** Kathryn Wilsterman.

**Funding acquisition:** Kathryn Wilsterman.

**Investigation:** Kathryn Wilsterman, Megan J Hemmerlein, Anna Isabel Bautista, Natalie M Báez-Torres, Kaylinn M Gosney, Kylie E Jewett, Ashley M Larson, Ellery L Myers.

**Methodology:** Kathryn Wilsterman, Megan J Hemmerlein, Anna Isabel Bautista, Natalie M Báez-Torres, Kaylinn M Gosney, Kylie E Jewett, Ashley M Larson, Ellery L Myers.

**Project administration:** Kathryn Wilsterman, Megan J Hemmerlein, Ashley M Larson.

**Visualization:** Kathryn Wilsterman, Megan J Hemmerlein.

**Writing – original draft:** Kathryn Wilsterman.

**Writing – review & editing:** Kathryn Wilsterman, Megan J Hemmerlein, Anna Isabel Bautista, Natalie M Báez-Torres, Kaylinn M Gosney, Kylie E Jewett, Ashley M Larson.

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
