## [Decision Letter · Decision Letter 0]

5 Mar 2025

Dear Dr. Wilsterman,

Thank you for submitting your manuscript to PLOS ONE. After careful consideration, we feel that it has merit but does not fully meet PLOS ONE’s publication criteria as it currently stands. Therefore, we invite you to submit a revised version of the manuscript that addresses the points raised during the review process.

We look forward to receiving your revised manuscript.

Kind regards,

Rupasri Ain, PhD

Academic Editor

PLOS ONE

Journal Requirements:

3. In the online submission form, you indicated that the data and scripts underlying the results presented in the study are available at https://github.com/mjhemmerlein/Peromyscus-Gestational-Atlas. Digital copies of all images shown or referenced in the paper can be obtained by contacting the corresponding author.

4. Please amend the manuscript submission data (via Edit Submission) to include author Kathryn Wilsterman.

5. Please amend your authorship list in your manuscript file to include author Kate Wilsterman.

6. Please remove all personal information, ensure that the data shared are in accordance with participant consent, and re-upload a fully anonymized data set.Note: spreadsheet columns with personal information must be removed and not hidden as all hidden columns will appear in the published file.Additional guidance on preparing raw data for publication can be found in our Data Policy (https://journals.plos.org/plosone/s/data-availability#loc-human-research-participant-data-and-other-sensitive-data) and in the following article:
http://www.bmj.com/content/340/bmj.c181.long.

Additional Editor Comments :

Dear Dr. Wilsterman,

Your manuscript entitled "A physiological and histological atlas of reproduction in the North American deer mouse (Peromyscus maniculatus)” has been reviewed by experts in the subject.

The reviewers were of the opinion that the manuscript contains important information that is likely to be of interest to other investigators.

However, they also identified a number of comments that require further attention, as indicated in the reviews appended below.

I encourage you to respond to the reviewers' comments and satisfy the concerns raised in both critiques.

If you do decide to resubmit, it is important that you pay attention to all the major and minor issues raised in the reviews, since I will once again be asking the same reviewing editors for their opinion on any revised manuscript.

Please note that any invitation to submit a revised manuscript should not be taken as any guarantee of acceptance or publication, and that the decision to

resubmit rests solely with the authors.

Thank you for choosing PLOS ONE, and we look forward to your revised manuscript.

Kind regards,

Rupasri Ain

Academic Editor

PLOS ONE

Reviewers' comments:

Reviewer's Responses to Questions

**Comments to the Author**

1. Is the manuscript technically sound, and do the data support the conclusions?

Reviewer #1: Yes

Reviewer #2: Yes

2. Has the statistical analysis been performed appropriately and rigorously?

Reviewer #1: Yes

Reviewer #2: Yes

3. Have the authors made all data underlying the findings in their manuscript fully available?

Reviewer #1: Yes

Reviewer #2: Yes

4. Is the manuscript presented in an intelligible fashion and written in standard English?

Reviewer #1: Yes

Reviewer #2: Yes

Reviewer #1: Manuscript Number: PONE-D-25-03756

A physiological and histological atlas of reproduction in the North American deer

mouse (Peromyscus maniculatus)

Wilsterman et al effective establishes the importance of North American deer mouse (Peromyscus maniculatus) as a model organism in multiple biological fields particularly in reproductive biology that focused on characterizing body temperature profiles across reproductive stages and generating a detailed histological atlas of placental development. Results show that body temperature can be used to diagnose copulation and pregnancy in deer mice. Moreover, authors found a unique organization and behaviours of trophoblast cells, especially at the maternal-fetal interface in the deer mouse. The objectives of the study were clearly setup and contributed to a detailed comprehensive understanding of reproduction in deer mouse. The manuscript is nicely design and conceived and I indeed enjoyed reading the paper. I have some points for revision which I feel to compulsory address and incorporate in the manuscript (see my comments below and – for the ease of the authors). There are English grammatical mistakes and needs to be revised throughout the manuscript

As mentioned in the study that male presence disrupted female body temperature rhythms. It is unclear that male presence influenced the female reproductive physiology, for instance: did authors examine female cortisol levels for stress? and any changes in breeding behaviour of females?

Moreover, on what basis author selected the gestational stages for placental analysis is not clearly justified. Are these stages were selected on earlier literature or studies??

Results show that cyclicity is not a reliable predictor of fertility in P. maniculatus, as only 2 cycling females became pregnant, and 7 were found to be non-cycling females.

Did male presence, stress or other environmental factors influenced cyclicity detection”

How frequently pseudo-pregnancy occurs in P. maniculatus.

Line 646. I suggest to move table 1 in results section from discussion

Line 598. Please change “in a other mammals to in other mammals”

Line 668. Change to “specific”

Line 611. Change to “are not clear to is not clear”

Line 654-655. Change to “Further physiological and histological studies are needed to clarify the roles of trophoblasts and uNK cells in vascular remodeling in deer mice”

Reviewer #2: Wilsterman et al have conducted have used remote temperature sensors to assay body temperature in wild caught Peromyscus maniculatus (deer mice) during mating, gestation, and lactation. In addition, they have conducted an immunohistological analysis of placental tissue. This research expands the model systems available for understanding mammalian implantation and gestation and highlights the similarities and differences between species. This work will be of interest to researchers working with Peromyscus and to the broader reproductive biology community.

My main criticism is that I would like to see more annotation on the figures that highlight the descriptive text. For instance, in supplemental figure 1 can you provide annotations that highlight the differences in the plots for cycling vs non-cycling females, or indicating the endometrial glands in figure 3 as referenced at line 354. Please apply this suggestion to all figures and not just these two examples.

In the results section, can you state how copulation was determined? I believe you annotate copulation retrospectively using the temperature data and that you don’t have precise evidence. Or are you using the increase in temperature in the rest phase (daylight?) as evidence of copulation? I use a lab stock of P. maniculatus (BW) in my research and I do vaginal gavage to look for sperm as an indication of copulation. I came to Peromyscus research from Mus, so I was used to “mice” mating in the dark period. I have found that the BW deer mice will mate in the light period as I find sperm when I do checks in the morning and the afternoon. A female that does not have sperm present in the morning gavage can have sperm present in the afternoon gavage.

In figure 2, please provide a code for your significance indicators.

At line 305, please indicate why you chose to start your analysis at e11.5.

In the figure 4 legend, please remove the reference to laminin data that isn’t shown.

**Do you want your identity to be public for this peer review?** For information about this choice, including consent withdrawal, please see our Privacy Policy

Reviewer #1: **Yes: ** Dr. Vinod Kumar

Reviewer #2: No

---

## [Author Response · Author response to Decision Letter 1]

21 Mar 2025

Please see attached "Response to Reviewers" document included the files.

---

## [Editor Report · Decision Letter 1]

6 Apr 2025

A physiological and histological atlas of reproduction in the North American deer mouse (Peromyscus maniculatus)

PONE-D-25-03756R1

Dear Dr. Wilsterman,

We’re pleased to inform you that your manuscript has been judged scientifically suitable for publication and will be formally accepted for publication once it meets all outstanding technical requirements.

Kind regards,

Rupasri Ain, PhD

Academic Editor

PLOS ONE

---

## [Editor Report · Acceptance letter]

PONE-D-25-03756R1

PLOS ONE

Dear Dr. Wilsterman,

I'm pleased to inform you that your manuscript has been deemed suitable for publication in PLOS ONE. Congratulations! Your manuscript is now being handed over to our production team.

Kind regards,

on behalf of

Dr. Rupasri Ain

Academic Editor

PLOS ONE